# Sim-to-Real Transfer via 3D Feature Fields for Vision-and-Language Navigation

**Zihan Wang**[1,2,3], **Xiangyang Li**[1,2], **Jiahao Yang**[1,2], **Yeqi Liu**[1,2,4], **Shuqiang Jiang**[1,2,3,4]
[1]Institute of Computing Technology, Chinese Academy of Sciences, Beijing
[2]University of Chinese Academy of Sciences, Beijing
[3]Institute of Intelligent Computing Technology, CAS, Suzhou  [4]Peng Cheng Laboratory, Shenzhen
zihan.wang@vipl.ict.ac.cn, lixiangyang@ict.ac.cn,
{jiahao.yang, yeqi.liu}@vipl.ict.ac.cn, sqjiang@ict.ac.cn

**Abstract:** Vision-and-language navigation (VLN) enables the agent to navigate to a remote location in 3D environments following the natural language instruction. In this field, the agent is usually trained and evaluated in the navigation simulators, lacking effective approaches for sim-to-real transfer. The VLN agents with only a monocular camera exhibit extremely limited performance, while the mainstream VLN models trained with panoramic observation, perform better but are difficult to deploy on most monocular robots. For this case, we propose a sim-to-real transfer approach to endow the monocular robots with panoramic traversability perception and panoramic semantic understanding, thus smoothly transferring the high-performance panoramic VLN models to the common monocular robots. In this work, the semantic traversable map is proposed to predict agent-centric navigable waypoints, and the novel view representations of these navigable waypoints are predicted through the 3D feature fields. These methods broaden the limited field of view of the monocular robots and significantly improve navigation performance in the real world. Our VLN system outperforms previous SOTA monocular VLN methods in R2R-CE and RxR-CE benchmarks within the simulation environments and is also validated in real-world environments, providing a practical and high-performance solution for real-world VLN. The code is available at https://github.com/MrZihan/Sim2Real-VLN-3DFF

**Keywords:** Vision-and-Language Navigation, 3D Feature Fields, Semantic Traversable Map

## 1 Introduction

Vision-and-Language Navigation (VLN) tasks [1, 2, 3, 4] require an agent to understand natural language instructions and move to the destination. In the continuous environment setting (*i.e.*, VLN-CE) [4], the navigation agent is free to traverse any unobstructed location with low-level actions (*e.g.*, turn left 15 degrees, turn right 15 degrees, or move forward 0.25 meters). These VLN agents are typically trained and evaluated within simulators [5, 6], equipped with panoramic RGB-D cameras or just a forward-facing RGB-D camera.

Upon receiving instructions, the VLN agent begins exploring, moving gradually to the destination to find the target, as shown in Figure 1. However, agents equipped with monocular cameras [7, 8] have a limited field of view (yellow sector in Figure 1), restricting their environmental perception. This limitation often leads to navigation failures, such as missing the target, exemplified by the *basketball* in Figure 1 (b). To enhance performance on benchmarks, most VLN agents [9, 10, 11, 12] use panoramic cameras (blue circular area in Figure 1), achieving nearly a 20% increase in navigation success rate. However, this trick has significantly hindered the deployment of VLN models on physical robots, primarily because most robots don't come equipped with panoramic RGB-D cameras due to their high cost and large size. Our goal is to design a VLN system that

8th Conference on Robot Learning (CoRL 2024), Munich, Germany.

enables robots with only a monocular camera to achieve near-panoramic perception, ensuring both practicality and high performance. In this way, two challenges must be addressed: **1)** identifying traversable areas around the robot, and **2)** representing semantic and spatial relationships within the panorama.

**Instruction:** *Go past the table with a laptop, find the basketball.*

Figure 1: The VLN models [7, 8, 13] equipped with a monocular camera have limited navigation success rates of less than 39% on the R2R-CE Val Unseen split. Most VLN models [9, 10] are trained and evaluated in the simulator [6] with the panoramic observation, achieving navigation success rates of over 57%, but hard to deploy on real-world robots.

For the first challenge, we propose a method to construct the semantic traversable map to identify traversable areas and navigable waypoints around the VLN robot, simplifying the action space and achieving obstacle avoidance in real-world navigation. Previous works [14, 9] use panoramic RGB-D images and multi-layer transformers to model spatial relationships and predict candidate waypoints. However, due to strict camera hardware requirements, these VLN models have not been successfully deployed and validated in real-world environments. In contrast, more practical VLN robots [15] use 2D laser scanners to create radial occupancy maps around the robot and employ a UNet [16] to predict navigable waypoints. This approach has two main drawbacks: **1)** The 2D laser scanner can only detect obstacles on a fixed-height plane, which significantly reduces real-world VLN performance [15]. **2)** The 2D laser scanner cannot identify obstacles with traversability such as stairs and doors. Using depth cameras for occupancy mapping, our solution can detect obstacles at varying heights. Furthermore, leveraging environmental layout knowledge acquired through pre-training, our semantic map predictor can forecast the obstacle layout beyond the current field of view. Additionally, the semantic map in our approach can also significantly identify traversable landmarks (*e.g.*, stairs and doors). All these advantages of the semantic traversable map enhance the navigation abilities of real-world robots.

For the second challenge, the 3D Feature Fields model [12] is utilized to represent the visual semantics of each waypoint to select the optimal one to move. During navigation, the 3D Feature Fields model [12] encodes 2D visual observations and projects them into 3D feature space via the depth map. Using volume rendering [17], the model can decode semantic representation of novel views from the feature fields and align them with the CLIP embeddings [18]. The 3D Feature Fields can generalize to unseen scenes, allowing real-time construction and dynamic updates. In our work, the feature fields constructed from historical observations can predict the novel view representations within the robot's panorama, significantly broadening the limited field of view of the monocular robot (69° HFOV and 42° VFOV), and improving navigation performance in the real world.

In this work, our main contributions include:

- We design a novel approach using a monocular RGB-D camera to construct the Semantic Traversable Map for predicting navigable waypoints and introduce the 3D Feature Fields for predicting novel view representations, endowing monocular VLN robots with panoramic perception capabilities.

- Our work provides a practical and high-performance solution for VLN sim-to-real transfer. Extensive experiments in the simulator and the real-world environments demonstrate the effectiveness of our approach.

## 2 Related Works

**Vision-and-Language Navigation.** Vision and Language Navigation (VLN) [1, 2, 4, 19, 20, 21, 10, 22, 23] has gained significant attention in recent years. Unlike object goal navigation [24, 25, 26, 27, 28, 29], VLN agents must understand more complex natural language instructions and navigate to the described destination. Early VLN research focused on discrete environments due to the computational demands of exploring large action spaces in continuous environments. In discrete environments like the Matterport3D simulator [5], a predefined navigation connectivity graph is used. Agents observe panoramic RGB and depth images and can teleport to nearby nodes in the graph, navigating from a starting node to a target node following natural language instructions. Despite efficient training and high navigation performance, these settings are impractical for real-world applications. To address this, the continuous environment setting (VLN-CE [4]) was introduced. In VLN-CE, agents freely navigate any unobstructed location in 3D environments using low-level actions (*e.g.*, turn left 15 degrees, turn right 15 degrees, or move forward 0.25 meters) in the Habitat simulator [6]. To overcome inefficiencies and poor performance in atomic action prediction, many works have designed subgoal or waypoint predictors to generate navigable waypoints, making VLN-CE more similar to discrete environments. By predicting several candidate waypoints, the agent selects the optimal one and executes atomic actions to move. This strategy significantly improves VLN-CE performance. However, these methods often rely on panoramic RGB-D images [15, 30, 14], limiting their applicability to real-world robots. In contrast, our work utilizes only a monocular RGB-D camera to construct a dynamic global semantic and occupancy map and then predicts the candidate waypoints, with better practicality.

**Sim-to-Real Transfer for VLN.** Anderson [15] makes the first attempt at VLN sim-to-real transfer with panoramic cameras, which propose a subgoal model based on a 2D laser scanner to identify nearby waypoints. He finds that sim-to-real transfer to an unseen environment is challenging with no prior navigation graph. The core issue is that the navigable candidate waypoints predicted by the subgoal model exhibit significant deviations compared to the pre-defined navigation graph. More recently, to leverage the powerful reasoning and generalization capabilities of large language models or multimodal models to assist in real-world VLN, VLMaps [31] and DiscussNav [32] achieve VLN by combining multiple large models or foundation models. However, the navigation performance of these zero-shot methods is very limited. Therefore, NaVid [7] attempts to fine-tune the large model [33] using a large amount of video-based VLN data to achieve superior performance for real-world VLN with only a monocular camera. The large models for step-by-step atomic action prediction incur significant computational costs and response delays. Our work aims to design a universal VLN framework for sim-to-real transfer with practicality, high performance, and low cost.

**Generalizable 3D Feature Fields.** The neural radiance field (NeRF) [17, 34] has gained significant popularity in various AI tasks, which predicts an image from an arbitrary viewpoint in a scene. However, the traditional NeRF methods with implicit MLP networks can only synthesize novel view images in seen scenes, which makes it difficult to generalize to unseen scenes and adapt to many embodied AI tasks. To avoid the difficulties of pixel-wise RGB reconstruction in unseen environments, GeFF [35] and HNR [12] attempt to encode 2D visual observations into 3D representations (called 3D feature fields) via the depth map. Using volume rendering [17], these models decode novel view representations from the 3D feature field and align them with open-world features (e.g., CLIP embeddings [18]). The 3D feature fields can generalize to unseen scenes, enabling real-time construction and dynamic updates, which assist with various embodied tasks. In this work, we utilize the HNR model [12] to provide panoramic perception capabilities for monocular robots.

## 3 Methods

### 3.1 Overview

**The baseline framework.** As shown in Figure 2, for most VLN-CE works [9, 10, 11], at each time step $t$, the agent observes panoramic RGB images $\mathcal{R}_t = \{r_{t,i}\}_{i=1}^{12}$ and depth images $\mathcal{D}_t = \{d_{t,i}\}_{i=1}^{12}$

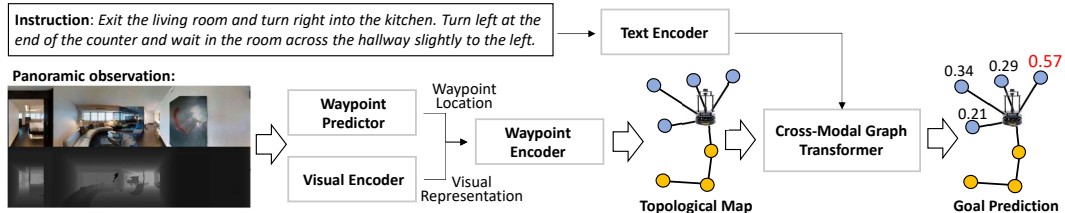

Figure 2: The VLN-CE model [9] with panoramic RGB-D observation.

surrounding its current location (*i.e.*, 12 view images with 30 degrees separation). Subsequently, a pre-trained waypoint predictor [14] takes the panoramic observation $\mathcal{R}_t$ and $\mathcal{D}_t$ as input and predicts the navigable waypoints around the agent. The VLN model typically encodes the visual observation of these waypoints with their location information (relative direction and distance) and then constructs a topological map. The Cross-Modal Graph Transformer [9, 20] is used to encode the topological map with the instruction and selects the optimal waypoint as the next goal to move. However, the pipeline above is challenging to implement for monocular robots due to its severely restricted field of view.

**The sim-to-real transfer framework.** As shown in Figure 3 and Figure 4, we design a new waypoint predictor based on the semantic traversable map, capable of predicting waypoints with only a monocular RGB-D camera, achieving nearly 360° coverage. Meanwhile, the 3D Feature Fields model [12] is used to predict visual representations of these waypoints even outside the field of view. Using Semantic Traversable Map and 3D Feature Fields, we successfully transfer several VLN-CE models with panoramic observation (ETPNav [9] and GridMM [11]) to real-world monocular robots, achieving superior performance.

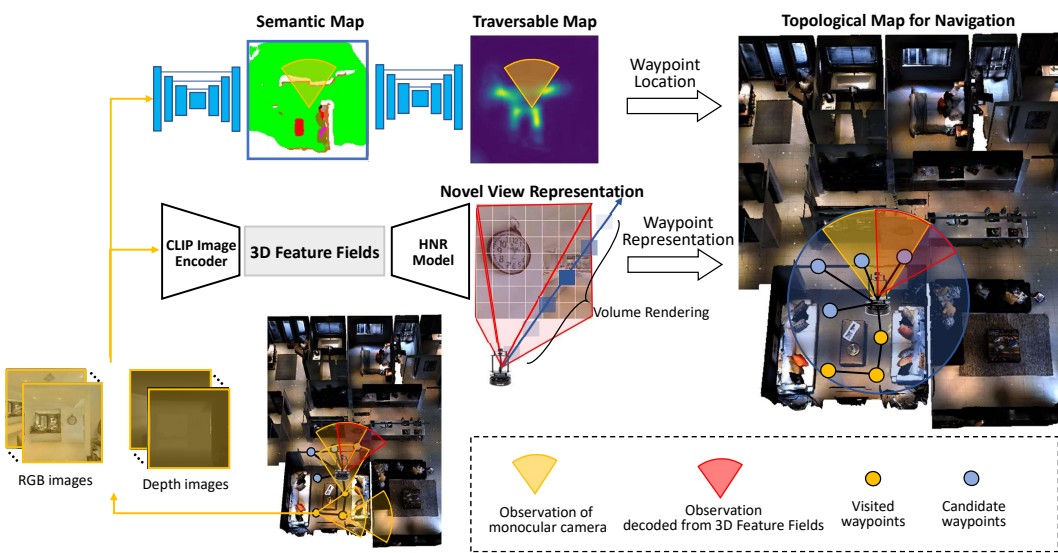

Figure 3: The sim-to-real transfer framework via semantic traversable map and 3D feature fields for vision-and-language navigation.

## 3.2 Semantic Traversable Map for Waypoint Prediction

**Ground-projecting the global map.** As shown in Figure 4, we design a waypoint predictor that generates the egocentric traversable map based on the global semantic map and occupancy map updated by monocular RGB-D observations. During navigation, following [36, 13], the model ground-projects depth observations to a global occupancy map $O_t \in \mathbb{R}^{h \times w \times 3}$ with classes $void$, $free$, and $occupancy$, where $h = w = 512$ with each pixel corresponding to $5cm \times 5cm$ region. Similarly,

the semantic segmentation is extracted by a pre-trained UNet model [13] and ground-projected to a global semantic map $S_t \in \mathbb{R}^{h \times w \times 27}$ with 27 classes. These global maps are accumulated and updated continuously during navigation.

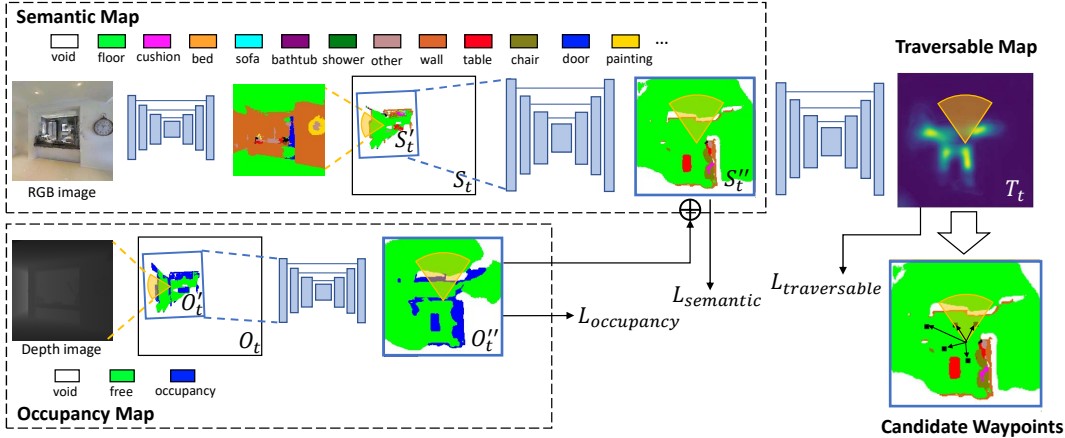

Figure 4: The framework of the waypoint predictor with semantic map and occupancy map [13].

**Generating the agent-centric subregion map.** To facilitate the prediction of a traversable map, the agent-centric $192 \times 192$ subregions $O_t'$ and $S_t'$ are cropped from the global semantic map $S_t$ and occupancy map $O_t$, respectively. To filter out noise in these subregion maps and fill in unobserved *void* areas as much as possible, two learnable UNet models [16, 13] are employed. The UNet models take the subregion maps $O_t'$ and $S_t'$ as inputs, concatenated with learnable positional embeddings $P \in \mathbb{R}^{192 \times 192 \times 16}$, and predict the refined semantic map $O_t'' \in \mathbb{R}^{192 \times 192 \times 3}$ and occupancy map $S_t'' \in \mathbb{R}^{192 \times 192 \times 27}$. These models are trained with the pixel-wise cross-entropy loss on the semantic and occupancy classes:

$$\mathcal{L}_{semantic} = \text{CrossEntropy}(S_t'', S_t^{gt}); \quad \mathcal{L}_{occupancy} = \text{CrossEntropy}(O_t'', O_t^{gt}) \tag{1}$$

The ground-truth semantic maps and occupancy maps are obtained following $CM^2$ [13] from the 3D semantic information in Matterport3D dataset [5].

**Predicting the traversable map.** Finally, the refined semantic map $S_t''$, occupancy map $O_t''$ and the learnable positional embedding are concatenated as the input $\in \mathbb{R}^{192 \times 192 \times (27+3+16)}$ for another UNet model, which outputs a probability distribution map of the traversable areas $T_t \in \mathbb{R}^{192 \times 192}$. The ground-truth traversable map is derived from the annotations of CWP [14]. CWP takes nodes from the navigable connectivity graph of the discrete R2R dataset [1] as the ground-truth waypoints. All these waypoints near the agent are used to construct 2D Gaussian distribution maps $\{G_k \in \mathbb{R}^{192 \times 192}\}_{k=1}^{K}$ centered at these waypoints with $\sigma = 5$. The ground-truth traversable map is obtained by combining these 2D Gaussian distributions into a single 2D mixture Gaussian distribution:

$$T_t^{gt} = \frac{\sum_{k=1}^{K} G_k}{K} \tag{2}$$

The mean squared error (*i.e.*, MSE loss) is used to optimize the prediction of the traversable map:

$$\mathcal{L}_{traversable} = \text{MSE}(T_t, T_t^{gt}) \tag{3}$$

**Predicting the candidate waypoints.** To obtain navigable waypoints from the traversable map, the map is segmented into 12 sections, each spanning 30 degrees from the center. The waypoint with the highest value in each section is selected, but waypoints in occupied regions of the occupancy map $O_t''$ are excluded. Ultimately, the Top-K waypoints with the highest probability values are chosen as the candidate waypoints for navigation.

### 3.3 3D Feature Fields for Panoramic Perception

After identifying candidate waypoints, a hierarchical neural radiance representation (HNR) model [12] is utilized to encode 3D feature fields and decode novel view representations within the

panorama. As shown in Figure 3, during navigation, the monocular RGB images are processed using a pre-trained CLIP-ViT-B/16 [18] model to extract fine-grained visual features and then mapped to 3D world positions using depth map and camera parameters.

**Volume rendering.** For decoding novel view representations, the HNR model generates a feature map $\mathbf{R} \in \mathbb{R}^{8 \times 8 \times 512}$ (which contains $8 \times 8$ subregions) for novel view by predicting subregion features through volume rendering in 3D feature fields. Specifically, the HNR model uniformly samples $N$ points along the ray from the camera position to the predicted subregion's center, search for k-nearest features of these points, and predicts the volume density $\sigma_n$ and latent vector $\mathbf{r}_n$ using an MLP network for each sampled point. These latent vectors are composited into a subregion feature $\mathbf{R}_{(u,v)}$ through volume rendering, as shown in Figure 3 and follows:

$$\mathbf{R}_{(u,v)} = \sum_{n=1}^{N} \tau_n (1 - \exp(-\sigma_n \Delta_n)) \mathbf{r}_n, \text{ where } \tau_n = \exp(-\sum_{i=1}^{n-1} \sigma_i \Delta_i) \tag{4}$$

$\tau_n$ represents volume transmittance, and $\Delta_n$ is the distance between adjacent sampled points. $\mathbf{R}_{(u,v)}$ represent the region feature at the $u$-th row and $v$-th column of the predicted feature map $\mathbf{R}$.

**Novel view representation.** Finally, a transformer-based decoder is used to aggregate the feature map $\mathbf{R} \in \mathbb{R}^{8 \times 8 \times 512}$ and generate the novel view representation $V \in \mathbb{R}^{1 \times 512}$, which is aligned with CLIP embedding of the novel view. At each step of navigation, via 3D feature fields and the HNR model, the VLN agent can perceive 12 views within the panorama. The forward-facing view feature is extracted from the monocular RGB image using the CLIP model, and the other 11 novel view features are predicted from the 3D feature fields. All these representations are input into the baseline VLN models in Section 3.1 for navigation.

## 4 Experiments

### 4.1 Experiment Setup

**Simulated environments.** We evaluate our approach on the R2R-CE [4] and RxR-CE [2] datasets in simulated environments [6]. **R2R-CE** [4] is collected based on the Matterport3D scenes [5] with the Habitat simulator [6]. The R2R-CE dataset includes 5,611 trajectories divided into the train, validation seen, validation unseen, and test unseen splits. Each trajectory has three English instructions, with an average path length of 9.89 meters and an average instruction length of 32 words. The camera's HFOV for R2R-CE is 90°. **RxR-CE** [2] is a larger multilingual VLN dataset containing 126K instructions in English, Hindi, and Telugu. It includes diverse trajectories in terms of length (the average is 15 meters), which is more challenging in continuous environments. The camera's HFOV for RxR-CE is 79°.

**Real-world environments.** We conduct all experiments using an Interbotix LoCoBot WX250 equipped with an Intel RealSense D435 camera to capture both depth and RGB images. The experiments are carried out using ROS Noetic in an unstructured lab environment unseen to all VLN models, these VLN models are run on a laptop with the GTX 1060 GPU. The lab environment is approximately 100 $m^2$, featuring movable walls and various pieces of furniture. We build five different house layouts, and each layout contains four types of rooms: *living room*, *kitchen*, *bathroom*, and *bedroom*. We annotate 20 data samples (trajectories and instructions) for each layout, resulting in 100 samples for real-world VLN. Details of these examples can be found in the appendix.

**Metrics.** There are several standard metrics [1] for evaluating the agent's performance in VLN-CE, including Navigation Error (NE), Success Rate (SR), SR given the Oracle stop policy (OSR), Success Rate weighted by normalized inverse Path Length (SPL).

**Training Details.** The predictor of the semantic traversable map is trained for 20K episodes on two RTX 3090 GPUs. Each episode consisted of a 15-step trajectory randomly sampled in the Habitat simulator. This model is optimized by training losses in Section 3.2. Our approach uses the pre-trained 3D Feature Fields model [12] to predict novel view representations within the panorama, without further training. The VLN models are trained using 4 RTX 3090 GPUs for 10K episodes on

the R2R-CE dataset and 20K episodes on the RxR-CE dataset. All parameters are initialized from the trained baselines (*i.e.*, ETPNav [9] and GridMM [11]).

## 4.2 Comparison on Simulated Environments

Table 1 and Table 2 represent the performance of our proposed approach compared with existing VLN models on the R2R-CE and RxR-CE datasets respectively. Overall, compared with other monocular VLN methods, our approach "ETPNav w/ 3D Feature Fields" achieves state-of-the-art results in most metrics, demonstrating the effectiveness of the proposed approach. As illustrated in Table 1, for both the val unseen split and test unseen split of the R2R-CE dataset, our model outperforms previous works (*e.g.*, WS-MGMap [8] and NaVid [7]) over 7% on navigation success rate (SR). It can be noted that our method does not show a significant advantage in the SPL metric. This is because the better SPL metric requires a shorter navigation path as well as a higher navigation success rate. However, constructing 3D Feature Fields requires the agent to conduct more exploration and observation, resulting in extra movements. Nevertheless, the excellent navigation success rate (SR) is still sufficient to demonstrate the superiority of our approach.

| Camera | Methods | Val Unseen | | | | Test Unseen | | | |
|---|---|---|---|---|---|---|---|---|---|
| | | NE↓ | OSR↑ | SR↑ | SPL↑ | NE↓ | OSR↑ | SR↑ | SPL↑ |
| Panoramic | Sim-2-Sim [30] | 6.07 | 52 | 43 | 36 | 6.17 | 52 | 44 | 37 |
| | CWP-CMA [14] | 6.20 | 52 | 41 | 36 | 6.30 | 49 | 38 | 33 |
| | CWP-RecBERT [14] | 5.74 | 53 | 44 | 39 | 5.89 | 51 | 42 | 36 |
| | GridMM [11] | 5.11 | 61 | 49 | 41 | 5.64 | 56 | 46 | 39 |
| | BEVBert [10] | 4.57 | 67 | 59 | 50 | 4.70 | 67 | 59 | 50 |
| | ETPNav [9] | 4.71 | 65 | 57 | 49 | 5.12 | 63 | 55 | 48 |
| Monocular | CM$^2$ [13] | 7.02 | 41.5 | 34.3 | 27.6 | 7.7 | 39 | 31 | 24 |
| | WS-MGMap [8] | 6.28 | 47.6 | 38.9 | 34.3 | 7.11 | 45 | 35 | 28 |
| | NaVid [7] | **5.47** | 49.1 | 37.4 | **35.9** | - | - | - | - |
| Monocular | GridMM w/ Feature Fields | 6.36 | 52.7 | 40.3 | 28.7 | 6.86 | 49.4 | 37.5 | 25.5 |
| | ETPNav w/ Feature Fields | 5.95 | **55.8** | **44.9** | 30.4 | **6.24** | **54.4** | **43.7** | **28.9** |

Table 1: Evaluation on the R2R-CE dataset.

## 4.3 Comparison on Real-world Environments

We conduct real-world experiments on four different types of scenes: *living room*, *bedroom*, *kitchen*, and *bathroom*. As shown in Table 3, the methods using 3D Feature Fields demonstrate more significant advantages in real-world environments than in simulators, outperforming previous methods. The Intel RealSense D435 camera equipped on our robot has an HFOV of 69° and a VFOV of 42°, which is much smaller than the 90° HFOV and VFOV used in the simulator. The minimal field of view severely restricts the performance of VLN models in real-world environments. Fortunately, the VLN methods using 3D Feature Fields overcome this difficulty well with their stronger panoramic perception capabilities.

| Camera | Methods | Val Unseen | | | |
|---|---|---|---|---|---|
| | | NE↓ | OSR↑ | SR↑ | SPL↑ |
| Panoramic | CWP-CMA [14] | 8.76 | - | 26.6 | 22.2 |
| | CWP-RecBERT [14] | 8.98 | - | 27.1 | 22.7 |
| | ETPNav [9] | 5.6 | - | 54.8 | 44.9 |
| Monocular | CM$^2$ [13] | 8.98 | 25.3 | 14.4 | 9.2 |
| | WS-MGMap [8] | 9.83 | 29.8 | 15.0 | 12.1 |
| | A$^2$Nav [37] | - | - | 16.8 | 6.3 |
| | NaVid [7] | **8.41** | 34.5 | 23.8 | **21.2** |
| Monocular | ETPNav w/ Feature Fields | 8.79 | **36.7** | **25.5** | 18.1 |

Table 2: Evaluation on the RxR-CE dataset.

| Methods | Living Room | | Bedroom | | Kitchen | | Bathroom | | All | |
|---|---|---|---|---|---|---|---|---|---|---|
| | OSR↑ | SR↑ | OSR↑ | SR↑ | OSR↑ | SR↑ | OSR↑ | SR↑ | OSR↑ | SR↑ |
| CM$^2$ [13] | 17.1 | 11.4 | 23.5 | 11.8 | 12.9 | 6.5 | 17.6 | 11.8 | 17.0 | 10.0 |
| WS-MGMap [8] | 22.9 | 14.3 | 29.4 | 23.5 | 22.6 | 12.9 | 17.6 | 11.8 | 23.0 | 15.0 |
| GridMM w/ Feature Fields | 51.4 | 40.0 | 41.2 | 29.4 | 48.4 | **41.9** | 41.2 | 23.5 | 47.0 | 36.0 |
| ETPNav w/ Feature Fields | **54.3** | **45.7** | **52.9** | **47.1** | **58.1** | 38.7 | **47.1** | **35.3** | **54.0** | **42.0** |

Table 3: Evaluation on the real-world environments.

## 4.4 Ablation Study

**3D Feature Fields.** As shown in Table 4, we analyzed the crucial role of 3D feature fields in our method. "Panorama w/o Depth feature" is our baseline method ETPNav [9] without the depth features, using only the CLIP features extracted from the panoramic RGB images. In the "w/o Feature Fields" setting, only the CLIP features from the forward-facing view are retained, with all other directions' features set to zero. This leads to a performance drop of over 20% on OSR, SR, and SPL metrics compared to the baseline, which shows that directly transferring the panoramic VLN model to a monocular camera setting is completely unfeasible. By constructing the feature fields (*i.e.*, "w/ Feature Fields"), the predicted novel view representations around the agent enhance the perception capability, leading to a 12% improvement in SR metrics compared to "w/o Feature Fields". Furthermore, to validate the capability of 3D Feature Fields to memorize and adapt to new environments, we draw on lifelong strategies from previous navigation works [38, 39]. Specifically, we use a group of 10 navigation examples within the same scene as a whole episode. During navigation in an episode, the feature fields constructed in the previous examples are preserved (but the traversable map is not preserved, as a 2D map cannot handle cross-floor scenarios across different examples), in this way, the previously constructed feature fields can support subsequent navigation and gain performance improvements on "w/ Feature Fields + Memory" setting.

**Traversable Map.** As shown in Table 5, we analyze the critical roles of various components in the Semantic Traversable Map. The performance of both "w/o Semantic map" and "w/o Occupancy map" is lower than that of the "Full pipeline", indicating that the semantic map and occupancy map are both crucial for predicting traversable areas. Especially when the semantic map is not available, the lack of identifiable landmarks with traversability (*e.g.*, doors and stairs) significantly compromises navigation performance. Additionally, the importance of learnable positional embeddings is also confirmed for predicting traversable maps using UNet. Removing these positional embeddings (*i.e.*, "w/o Positional embeddings") significantly reduces the UNet's spatial understanding from an agent-centric view, thereby impairing navigation performance.

| Settings | NE↓ | OSR↑ | SR↑ | SPL↑ |
|---|---|---|---|---|
| Panorama w/o Depth feature | 4.84 | 62.9 | 56.1 | 47.1 |
| w/o Feature Fields | 6.81 | 42.4 | 32.9 | 23.1 |
| w/ Feature Fields | 5.95 | 55.8 | 44.9 | 30.4 |
| w/ Feature Fields + Memory | 5.88 | 57.4 | 45.8 | 32.1 |

| Settings | NE↓ | OSR↑ | SR↑ | SPL↑ |
|---|---|---|---|---|
| w/o Semantic map | 6.21 | 50.6 | 40.1 | 25.4 |
| w/o Occupancy map | 6.08 | 53.9 | 42.7 | 28.6 |
| w/o Positional embeddings | 6.13 | 52.4 | 42.3 | 27.9 |
| Full pipeline | 5.95 | 55.8 | 44.9 | 30.4 |

Table 4: Ablation study for the 3D Feature Fields on R2R-CE Val Unseen split.

Table 5: Ablation Study for the Semantic Traversable Map on R2R-CE Val Unseen split.

## 5 Conclusion

In this work, based on the proposed semantic traversable map and 3D feature fields, we enable VLN robots with monocular cameras to have panoramic perception capabilities, which help various VLN models achieve high-performance sim-to-real transfer. We verify the significant role of panoramic perception for vision-and-language navigation, as well as the feasibility of traversable map and 3D feature fields to enhance perceptual capabilities, which may help a broader range of embodied tasks.

**Limitations.** Since both the semantic traversable map and the 3D feature fields rely on information provided by past observations, the agent tends to engage in more exploration to gather sufficient environmental observations before making navigation decisions, which results in longer path length and a lower SPL metric. In the future, we are looking to improve the prediction ability of the traversable map and 3D feature fields for unobserved areas, thereby reducing extensive exploration.

**Acknowledgments**

This work was supported in part by the National Natural Science Foundation of China under Grants 62125207, 62102400, 62272436, and U23B2012, in part by Beijing Natural Science Foundation under Grant L242020, in part by the National Postdoctoral Program for Innovative Talents under Grant BX20200338.

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

# Appendix

## A Presentation of the Real-world VLN

Figure 5 shows the lab environment for the real-world navigation. The left part presents the arrangement of various types of rooms, and the right part presents the visualization of a house layout.

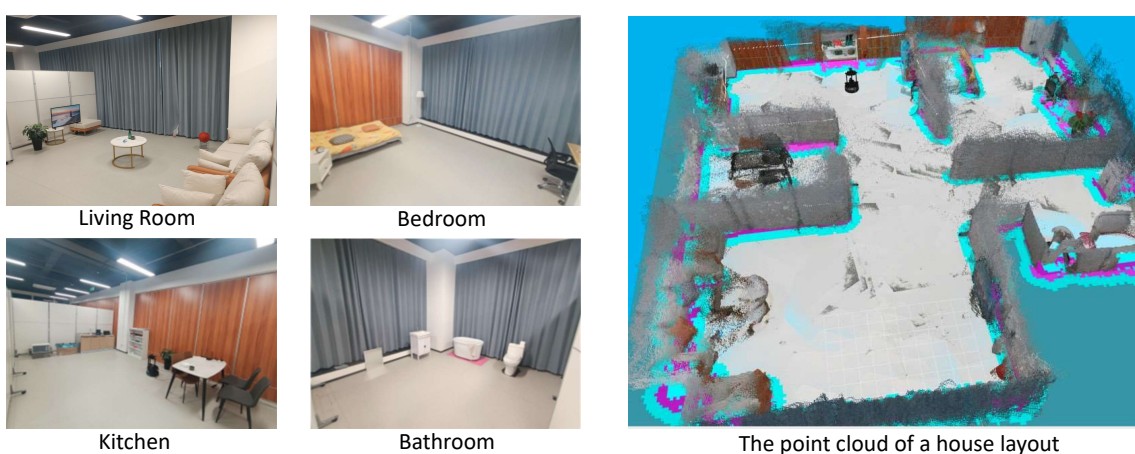

Figure 5: The arrangement and layout of the real-world lab environment.

Figure 6 shows two examples of the Interbotix LoCoBot navigating in the real-world environment. The robot can identify landmarks mentioned in the language instructions and successfully avoid obstacles during navigation.

**Instruction:** Go past the green plant into the bathroom, and stop in front of the toilet.

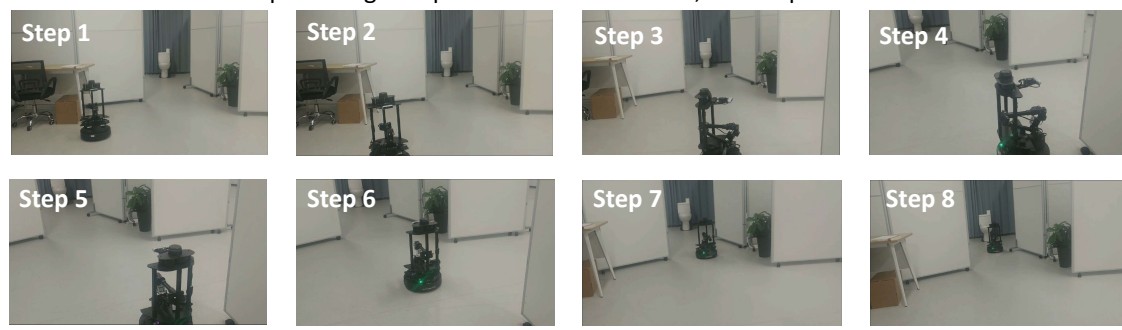

**Instruction:** Pass by the table and stop in front of the green plant next to the basketball.

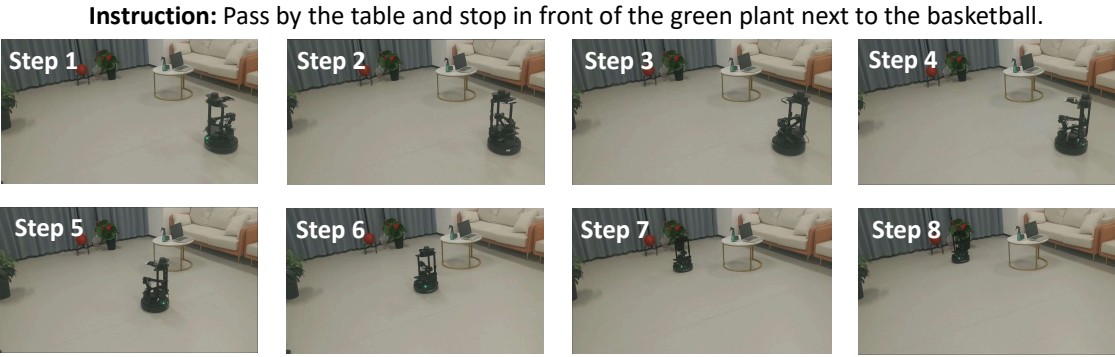

Figure 6: Examples of the vision-and-language navigation in the real-world environment.

# B  Visualization of the Semantic Traversable Map

Figure 7 and Figure 8 illustrate the construction and updating of the semantic map and traversable map during navigation. Both maps are agent-centered to better predict candidate waypoints. In the traversable map, warmer color indicates higher traversability probabilities, while the square black spots in the semantic map represent predicted navigable waypoints. In step 1 of navigation, the VLN model executes a 360-degree rotation to construct a more comprehensive semantic map and 3D feature fields. In the subsequent steps, the agent can only observe the forward-facing view. These examples illustrate that our approach can predict high-quality semantic maps of the environment, the traversable map can identify candidate waypoints well and achieve obstacle avoidance.

**Instruction**: Turn around, walk out the door on the right, and walk across the hall into the bedroom on the left.

Figure 7: (1/2) The visualization of the RGB observation, semantic map with candidate waypoints, and traversable map during navigation.

**Instruction**: Exit the room. Turn left and walk straight toward the pool. Wait near the pool.

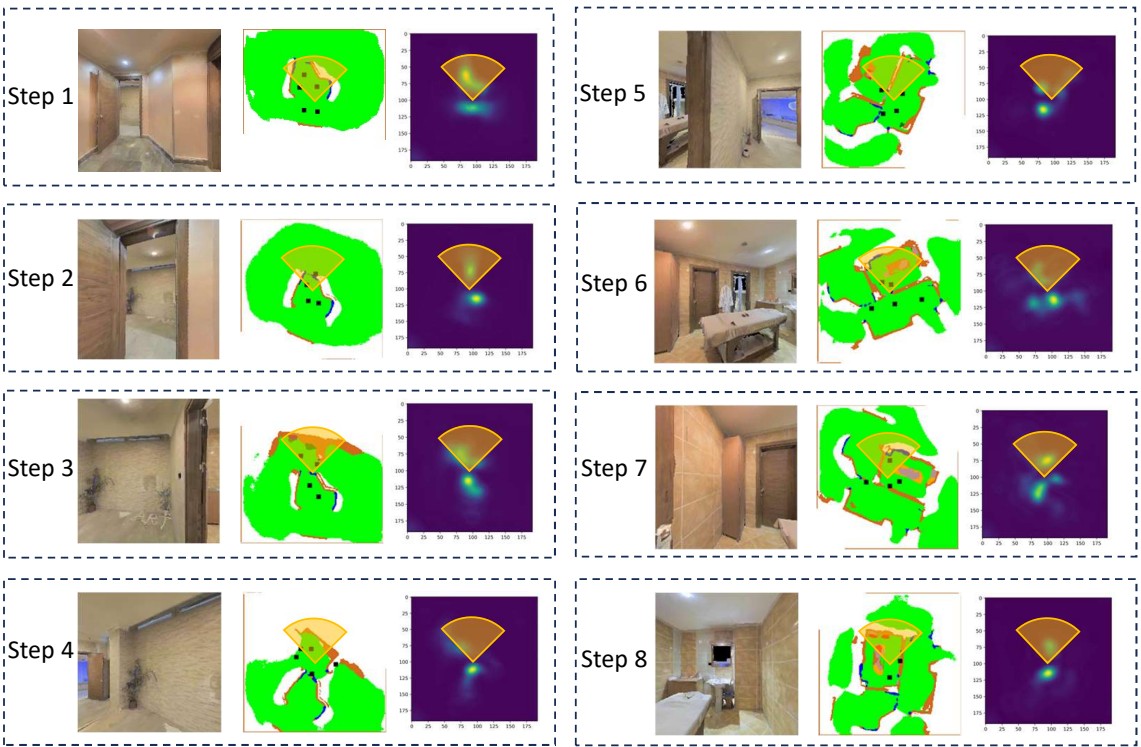

Figure 8: (2/2) The visualization of the RGB observation, semantic map with candidate waypoints, and traversable map during navigation.

