# OpenReview forum: "Sim-to-Real Transfer via 3D Feature Fields for Vision-and-Language Navigation"
_robot-learning.org/CoRL/2024/Conference — CoRL 2024_

### Official Review · Reviewer_yffA · 2024-07-15
**seems like a good work but not in my expertise**

**Originality:** 3
**Technical Quality:** 4
**Clarity Of Presentation:** 4
**Potential Impact:** 3
**Recommendation:** 3
**Confidence:** 1

**Review:**

The work proposes several techniques to deploy navigation robots with monocular RGBD camera and algorithmatically given panoramic traversability perception and semantic understanding. The techniques include predicting semantic traversable map for waypoint prediction and using 3D feature fields for panoramic perception. Training neural networks to do these allows real robots with only monocular limited-view RGBD camera to gain panoramic map and semantics and also allow sim-to-real.

Strengths:
- the proposed problem is a challenging yet important problem to address. The idea to build panoramic map and semantic understanding given only RGBD cameras is very important to the field.
- the proposed method has decent technical difficulty and novelty to the best of my judgement. The idea of using 3D generalizable feature field for potential sim-to-real transfer seems valid, interesting, and important to the field.
- experiments show clearly better performance than the state-of-the-art methods.
- ablation studies and analysis are also presented in a comprehensive and informative manner.

Weaknesses;
- mostly leveraging proposed technical components from prior work.
- longer navigation required

**Quality Of The Limitations Section:**

3

**Questions For Rebuttal:**

- how to address the longer navigation steps?

**Robotics Focus:**

4

**Summary Of Paper:**

The work proposes several techniques to deploy navigation robots with monocular RGBD camera and algorithmatically given panoramic traversability perception and semantic understanding.

**Summary Of Recommendation:**

not in my field of expertise, but looks like a good paper with decent story, method, and sufficient experiments to support

---

### Official Review · Reviewer_QJ8S · 2024-07-18
**The usage of 3D neural fields to increase perception is technically sound, and the results support it as a SOTA level performance work. However, I believe it uses many off-the-shelf models designed directly for the simulator[9, 20, 12], which could lead to overfitting to the simulators. Nevertheless, the real-world experiments support that it can work in the real world, although the trajectories are relatively short. Overall, I am led to a positive conclusion.**

**Originality:** 3
**Technical Quality:** 3
**Clarity Of Presentation:** 3
**Potential Impact:** 3
**Recommendation:** 3
**Confidence:** 4

**Review:**

Most of the paper is well-written with good illustrations, and the topic of panoramic and monocular observation is interesting. The benchmark experiments demonstrate the good performance of this paper.
But I have some concerns:
1.While I understand that the authors use a Cross-Modal Graph Transformer for waypoint selection, I would like to see a discussion or experiment about using feature neural fields with these methods.
2.I believe a very important experiment is to compare with a method where the robot rotates at each waypoint. This baseline can directly reflect the real performance of this method.
3.Many techniques rely on noiseless depth maps and odometry. I would like to know the authors’ comments on using these techniques in the real world.

**Quality Of The Limitations Section:**

2

**Questions For Rebuttal:**

See review part

**Robotics Focus:**

4

**Summary Of Paper:**

This paper proposes leveraging a traversable map and a 3D feature map for waypoint prediction and selection, respectively. By building the temporal map, the agent gains enhanced perception of indoor scenes. These improvements enable the agent to more successfully navigate and search indoor environments.

**Summary Of Recommendation:**

This paper is technically sound, and the experiments support most of the claims. I have some concerns about the experiments and real-world implementations, which are listed above. Overall, I am inclined to be positive if the authors can address my concerns.

---

### Official Review · Reviewer_U8bM · 2024-07-28

**Originality:** 3
**Technical Quality:** 4
**Clarity Of Presentation:** 3
**Potential Impact:** 3
**Recommendation:** 3
**Confidence:** 4

**Review:**

## Strength
- There is a novelty of extracting panoramic visual features using NeRF under narrow field-of-view conditions.
- The proposed method is well-explained. In particular, Figures 3 and 4 are clear and greatly aid in understanding the approach.
- Experiments are conducted across multiple benchmarks and environments, with appropriate baseline methods used for comparison.
- In sim2real experiments, the proposed method significantly outperforms baseline approaches, achieving scores comparable to simulation results.
- The ablation study appropriately analyzes contributions related to the method.
- The videos included in the supplementary materials are very effective in understanding the real-world experimental setup.
- The limitations section appropriately discusses the drawbacks of the proposed method, including potential future research directions.

## Weakness
- In the related work section,  in particular, the discussion of Vision-and-Language Navigation is insufficient. For example, highly relevant papers (e.g., [1], [2]) to this research are not cited.
- While the proposed method focuses on improving success rates, one concern is the apparent disregard for SPL (Success weighted by Path Length), which is an important metric in VLN tasks. If SPL can be ignored, the authors should discuss how success rates would be affected by using heuristic methods (e.g. an exhaustive search method). Furthermore, given that path length is sacrificed, the modest improvements - only 6 points on the R2R-CE dataset and only 1.7 points on the RxR-CE dataset - warrant further discussion.

[1] Hong, Y., Zhou, Y., Zhang, R., Dernoncourt, F., Bui, T., Gould, S., & Tan, H. Learning Navigational Visual Representations with Semantic Map Supervision. In Proceedings of the IEEE/CVF International Conference on Computer Vision, pages 3055-3067, 2023.
[2] Wang, H., Liang, W., Van Gool, L., & Wang, W. Dreamwalker: Mental Planning for Continuous Vision-Language Navigation. In Proceedings of the IEEE/CVF International Conference on Computer Vision, pages 10873-10883, 2023.

## Minor Comments
- Is the absence of bold formatting for the highest values in Tables 3, 4, and 5 intentional?
- Is the line to the right of NE only in Table 4 intentional? If it's because of improvement with memory, this should be explicitly stated.
- Are the two arrows next to SPL in Table 2 a typo?
- Lines 265-266 are questionable, as the scores for the main metrics have decreased.
- In equation (4), aren't $u$ and $v$ undefined?

**Quality Of The Limitations Section:**

3

**Questions For Rebuttal:**

Please review the above weaknesses. Below is an additional question.

- If constructing the 3D feature field results in longer paths, wouldn't it be just as effective to capture multiple viewpoints with the monocular camera and create a panoramic image? The authors should verify this with experimental results.

**Robotics Focus:**

4

**Summary Of Paper:**

This study proposes a Vision-and-Language Navigation method based on images captured by a monocular camera. The proposed method utilizes NeRF, enabling consideration of objects outside the view of the camera. The method is evaluated on two simulator-based benchmarks and real-world experiments. Results indicate that the proposed approach achieves a higher task success rate compared to existing monocular camera-based models.

**Summary Of Recommendation:**

This paper, as mentioned above, shows some concerns regarding the assumptions of the proposed method and contains several notation errors. However, the novelty of using NeRF to compensate for the limitations of a monocular camera is valuable, and this is supported by the experimental results. Moreover, the proposed method's effectiveness is reported across multiple benchmarks, which lends high credibility to the validity of its novelty. Therefore, my decision is a "weak accept."

---

### Decision · Program_Chairs · 2024-09-04

**Decision:**

Accept

**Comment:**

# Strengths
1. The idea of using a 3D generalizable feature field for sim-to-real transfer is interesting.
1. Experiments are conducted across multiple benchmarks.
1. The real-world demonstrations showcase the effectiveness of the approach.
1. Most of the paper is well-written and easily comprehensible.

# Weaknesses
1. Comparison experiments are insufficient. For example, the proposed method should be compared with a simple baseline where the robot rotates at each waypoint.
1. The paper focuses on improving some of the standard metrics, sacrificing other important metrics.
1. The references are not comprehensive.

### Post-rebuttal comment
The reviewers agree in their recommendation to accept the paper. I agree with their consensus.